# Effect of Moisture Content Difference on the Analysis of Quality Attributes of Red Pepper (*Capsicum annuum* L.) Powder Using a Hyperspectral System

**DOI:** 10.3390/foods11244086

**Published:** 2022-12-17

**Authors:** Ji-Young Choi, Jeong-Seok Cho, Kee Jai Park, Jeong Hee Choi, Jeong Ho Lim

**Affiliations:** 1Food Safety and Distribution Research Group, Korea Food Research Institute, Wanju 55365, Republic of Korea; 2Smart Food Manufacturing Project Group, Korea Food Research Institute, Wanju 55365, Republic of Korea

**Keywords:** red pepper powder, hyperspectral imaging, multivariate analysis, moisture adjustment

## Abstract

The variety of characteristics of red pepper makes it difficult to analyze at the production field through hyperspectral imaging. The importance of pretreatment to adjust the moisture content (MC) in the process of predicting the quality attributes of red pepper powder through hyperspectral imaging was investigated. Hyperspectral images of four types of red pepper powder with different pungency levels and MC were acquired in the visible near-infrared (VIS-NIR) and short-wave infrared (SWIR) regions. Principal component analysis revealed that the powders were grouped according to their pungency level, color value, and MC (VIS-NIR, Principal Component 1 = 95%; SWIR, Principal Component 1 = 91%). The loading plot indicated that 580–610, 675–760, 870–975, 1020–1130, and 1430–1520 nm are the key wavelengths affected by the presence of O-H and C-H bonds present in red pigments, capsaicinoids, and water molecules. The R^2^ of the partial least squares model for predicting capsaicinoid and free sugar in samples with a data MC difference of 0–2% was 0.9 or higher, and a difference of more than 2% in MC had a negative effect on prediction accuracy. The color value prediction accuracy was barely affected by the difference in MC. It was demonstrated that adjusting the MC is essential for capsaicinoid and free sugar analysis of red pepper.

## 1. Introduction

Red pepper (*Capsicum annuum* L.) is a single crop belonging to the Solanaceae family. It has a spicy taste and red color [1] and it is often dried and processed into a powder and used as a spice for food additives [2]. Preference for the quality of red pepper is ultimately determined by the taste components (mixed with spicy, sweet, and other flavor components) contained in red pepper powder. Homologs of capsaicinoids, which are components of hot pepper, include capsaicin, dihydrocapsaicin, nordihydrocapsaicin, and glucose and fructose, which are reducing sugars, and are particularly closely related to the overall preference of red pepper powder. In particular, sweet flavor is negatively correlated with capsaicin content and stinging pain [3].

It is cultivated in different varieties and even in the same variety, and the capsaicinoid and sugar contents differ depending on the cultivation conditions, such as sunlight, precipitation, soil characteristics, or the difference in harvest time [4]. The survey report of the Consumers Federation of Korea (2013) noted that 80% of consumers responded that a label on the taste of red pepper powder is necessary, which affects product purchases. Therefore, real-time quality monitoring is required to label objective information on products [5]. High-performance liquid chromatography (HPLC) and gas chromatography/mass spectrometry (GC/MS) have been used to measure the content of capsaicinoid in red pepper [6,7,8,9]. However, these methods have some disadvantages as they are time-consuming, destructive, and lack capable real time detection systems. Among alternative methods, hyperspectral imaging (HSI) technology, which combines spectroscopy and cameras, can simultaneously provide spectral and spatial information regarding the external and internal qualities of agricultural products and are advantageous as they are fast, non-destructive, and cost-effective [10,11]. To enhance the applicability of HSI, chemometric methods such as principal component analysis (PCA) and partial least squares (PLS) regression are widely used for spectral analysis of foods with complex characteristics, because they offer better flexibility in conditions such as multicollinearity and when the number of variables exceeds the number of samples [12]. 

Previously, various spectroscopic trials and chemometrics were performed to analyze quality characteristics including capsaicinoids, free sugars, and moisture content of red pepper and red pepper powder [13,14,15]. Because the water content and particle distribution of the powder affect the light penetration depth and reflective ability, which influence spectroscopic signals such as any physical interference and chemical signals [16,17,18], it has been conjectured that ensuring uniformity can improve the measurement accuracy of components such as capsaicinoid in red pepper powder [19]. Compared with sieving to make the particle size of red pepper powder uniform, it is practically difficult to apply the manufacturing process to ensure that the water content is the same in the field. Therefore, by confirming the prediction accuracy according to the range of the difference in moisture content between samples, no previous study has shown the need for moisture control in the spectroscopic analysis of red pepper powder or established the moisture distribution conditions for sample preparation.

In this study, the moisture content of red pepper powder with different levels of spiciness produced in Gochang-gun, Shintaein-eup, Gwanchon-myeon, and Jeongeup-si was adjusted to 7, 8, 9, 10, 11, and 12%, respectively. By extracting Vis-NIR (400–1000 nm) and SWIR (900–1700 nm) image spectrum information and performing multivariate analysis, the capsaicinoid content, free sugar content, and color prediction accuracy of red pepper powder were determined according to the range of moisture content difference (7–8%, 7–9%, 7–10%, 7–11%, and 7–12%). It was hypothesized that this process would be able to prove the extent of which the moisture content difference has a high reliability for each quality prediction model. This study provides a basis for application in the field of red pepper powder production by overcoming the limitations of hyperspectral image analysis, which is strongly influenced by the bonding of water molecules. It can be a useful reference for determining the range of moisture content in samples in hyperspectral analysis studies of various agricultural foods, as well as red pepper powder.

## 2. Materials and Methods

### 2.1. Sample Preparation

Red peppers produced in Gochang-gun (GC), Sintaein-eup (ST), Kwanchon-myeon (KC), and Jeongeup-si (JU) regions, Jeollabuk-do, Korea, in 2021 were purchased as samples. Red peppers were ground after hot air drying (50–60 °C), and the particle size of the red pepper powder was uniformly prepared with a particle size of 425–850 μm using a standard sieve. Samples were prepared based on the particle size of red pepper powder for seasoning, which is most commonly used in Korea, according to Korean Industrial standards (KS). To ensure that the moisture content of each sample was 7%, 8%, 9%, 10%, 11%, and 12%, the following process was performed. KS presents less than 13% as the appropriate moisture content of red pepper powder, and the average moisture content of red pepper powder sold on the market is 7–12%.

First, the moisture contents of the GC, ST, KC, and JU powder samples were measured using the atmospheric pressure drying method in a drying oven. The samples were dried at 100 °C for 4 h, and the moisture content was calculated using the weight differences before and after drying. The moisture contents of the GC, ST, KC, and JU were 11.12%, 11.36%, 10.12%, and 11.09%, respectively. To adjust the initial moisture content to 12%, it was necessary to seal the samples in plastic bags and humidify by spraying additional 10.75 mL, 7.81 mL, 22.57 mL, and 11.08 mL of water on 950 g of GC, ST, KC, and JU samples.

75 g of the red pepper powder whose moisture content was adjusted to 12% was dried in a dry oven set at 55 °C, and the weight of the red pepper powder was measured every 15 min. A graph was prepared as shown in Figure 1. The moisture content was calculated as the change between the initial weight and the weight after drying, and red pepper powder samples with moisture contents of 7%, 8%, 9%, 10%, 11%, and 12% were prepared. According to the production area and moisture content of the samples, Gochang samples were GC7, GC8, GC9, GC10, GC11, and GC12, Shintaein samples were ST7, ST8, ST9, ST10, ST11, ST12. Kwanchon samples were KC7, KC8, KC9, KC10, KC11, and KC12 and Jeongeup samples were named JU7, JU8, JU9, JU10, JU11, and JU12.

### 2.2. Determination of Quality Indicators

To analyze the capsaicinoid and free sugar content of the samples, pretreatment was required to make the particle size uniform. The samples were finely ground using a food mixer (SNSG-1002SS, Hanil Electric, Seoul, Korea), filtered through a 30 mesh sieve (pore size, 0.6 mm), and then used for analysis.

#### 2.2.1. Moisture Content Measurement and American Spice Trade Association (ASTA) Color

The moisture content of the red pepper powder was measured by drying for 6 h in a vacuum oven dryer (OV-11, Jeio Tech, Daejeon, Republic of Korea) set at 70 °C, according to ASTA analytical method 2.1. The ASTA color value measurement method was based on AOAC official method 971.26, and acetone was filled in 0.1 g of the sample, shaken for 1 min, and left in the dark for 16 h to prepare a test solution. The absorbance of the test solution was measured at 460 nm using a UV spectrophotometer (Thermo Fisher Scientific, Vantaa, Finland), and the results were substituted into the equation below to calculate the ASTA color value.
(1)$$\mathrm{ASTA}\;\mathrm{value}=\frac{A\times 16.4}{W}$$
A: absorbance at 460 nm; W: sample weight (g).

#### 2.2.2. Capsaicinoid

Capsaicin and dihydrocapsaicin contents were analyzed by referring to the methods of Ku et al. [20] and Namgung et al. [21]. The extraction method for capsaicinoid analysis was as follows: Methanol (10 mL) and a boiling chip were added to 2 g of the sample and heated on a dry heating block (MaXtable H10, Daehan, Incheon, Korea) set at 90 °C for 1 h, and then cooled to room temperature. The extract was filtered with Whatman No. 1 and then filtered again with a 0.2 μm syringe filter. Capsaicinoid content was analyzed using an HPLC system (Agilent 1260 infinity Ⅱ, Agilent Technology, Santa Clara, CA, USA). An XTerraTMRP18 (5 μm, 4.6 × 150 mm id., Waters, Milford, MA, USA) column was used, and the mobile phase (A: acetic acid, B: acetonitrile) was applied in a gradient method (A: B = 60:40, 38:62, and 20:80) at a rate of 1 mL/min. The column temperature was set at 35 °C and the injection volume was 10 μL. A variable-wavelength detector was used, and the absorbance was measured at 280 nm. Capsaicin and dihydrocapsaicin were used as standards to prepare calibration curves.

#### 2.2.3. Free Sugar

The free sugar content of the red pepper powder was analyzed by high-performance liquid chromatography (HPLC, Agilent 1260 infinity Ⅱ, Agilent Technology, CA, USA). 40 mL of 80% ethanol was added to 2 g of the sample, extracted for 1 min with a vortex mixer, filtered through a 0.2 μm membrane filter, and 20 μL was injected into the 1260 Ⅱ Infinity HPLC-Refractive Index (RI) detector for analysis. Fructose, glucose, and sucrose (Sigma-Aldrich, St. Louis, MO, USA) dissolved in 80% ethanol were used as the standards. For the mobile phase, a solvent mixture of acetonitrile and water at a ratio of 75:25 (*v*/*v*) was separated in the isocratic mode at a flow rate of 1 mL/min. The column temperature was set to 30 °C, and the temperature of the RI detector was set to 35 °C. All analysis processes were performed by referring to the methods of Ku et al. [2].

#### 2.2.4. Statistics Analysis

All experimental measurements of 24 samples were performed three times, and the results are presented as means and standard deviations (*n* = 72, mean ± SD). The results were analyzed by ANOVA and Duncan’s multiple range test (*p* < 0.05) using the SPSS software package (version 20, IBM SPSS Statistics, Inc., Chicago, IL, USA).

### 2.3. Hyperspectral Image Analysis

#### 2.3.1. Hyperspectral Image Acquisition and Data Extraction

Hyperspectral images in the VIS-NIR region (400–1000 nm) were acquired using the line scan method (pushbroom) using a SPECIM FX10 spectrometer (Spectral Imaging Ltd., Oulu, Finland) equipped with three halogen light sources. It was operated by obtaining the reflection intensity from the sample, and image data with a spectral resolution of 1.3 nm were acquired for a total of 448 bands. A white plate made of polytetrafluoroethylene and the sample were scanned together, and the acquired image was normalized using the IDL Virtual Machine Application program (8.8.0, L3Harris Geospatial, Boulder, CO, USA).

HSI data of the red pepper powders were acquired using an ImSpector N17E (Specim, Spectral Imaging Ltd., Oulu, Finland) in the short-wave infrared (SWIR) region, 900–1700 nm. The light source consisted of two halogen lamps (1400 nm long-pass filter). The system consisted of an NIR camera with an indium gallium arsenide (InGaAs) sensor operated in reflectance mode with line-by-line scanning (pushbroom) to obtain intensity images at 5 nm intervals through a 30 µm slit (256 images per scene). A white plate was used as the reference material and was scanned before each sample was scanned. The samples were scanned line-by-line along the *Y*-axis and moved along the *X*-axis to obtain a three-dimensional (3D) hypercube containing both spatial and spectral information.

The powder (3.5 g) was placed in a transparent Petri dish (5 cm diameter) and spread flat to cover the bottom of the Petri dish. To reduce the diffuse reflection that may have been caused by the particle surface, the surface of the sample was compressed with a presser to make it as level as possible. Fifty hyperspectral images were acquired per sample for a total of 1200 images. All the hyperspectral imaging systems were operated using Microsoft Windows. To obtain the necessary information from the acquired images, image spectrum data for the inner area of the Petri dish were obtained using the region of interest function of the ENVI (version 5.4, Exelis Visual Information Solutions, Boulder, CO, USA) program.

#### 2.3.2. Chemometrics

Chemometrics is a method of high-level interpretation of one-dimensional data obtained through chemical analysis using computers, mathematics, and statistics and was used in this study to link quality-related factors and measurement technology. Multivariate statistical analysis consists of unsupervised learning, which finds data patterns or relationships between data when the characteristics of the data are unknown, and supervised learning, which predicts results by finding the optimal model by learning through an algorithm set with input and output values [22].

In this study, principal component analysis (PCA), a representative unsupervised learning method, was performed to visualize the overall clustering tendency according to the sourness and moisture content of the red pepper powder samples. Two-dimensional and three-dimensional PCA score plots were derived from the spectral data in the 400–1000 nm and 900–1700 nm regions. As the number of principal components increases, overfitting occurs, and the reliability of the predictive model decreases [23], so the maximum principal component was set to 7. Principal component analysis was performed using Unscrambler statistics program (version 10.5, CAMO, Trondheim, Norway).

To predict capsaicinoid content, partial least squares regression (PLSR) analysis, a supervised learning method, was attempted. The PLS statistical method combines the functions of principal component analysis and multiple regression analysis and aims to predict the independent variable by expressing the relationship between the predictor variable X (spectral data) and the independent variable Y (measured capsaicinoid content) in a linear model [24]. The predicted value of Y was calculated using the following equation:(2)Y=βX+b 
β: vector of regression coefficient; b: model offset.

The PLS model showed more stable characteristics than the principal component model, considering only the independent variables. Of the total spectral data, 70% were used to develop the calibration model, and the remaining 30% were used for testing to verify the developed model. To evaluate the performance of all developed PLS models, the coefficient of determination (R_c_^2^) in the calibration model, coefficient of determination (R_v_^2^) in the cross-validation model, root mean square error of calibration (RMSEC), cross-validation model, and root mean square error of validation (RMSEV) value were considered. Table 1 shows the PLS model names developed in this study and the data samples (spectral and physicochemical data) inserted into each model. The entire model developed using samples with uniform moisture content was named Model A, and the entire model developed with samples having different moisture contents was named Model B.

## 3. Results and Discussion

### 3.1. Quality Indicators Analysis and Correlation between Physicochemical Properties

Table 2 shows the analysis results of physicochemical characteristics of red pepper powder. The moisture content showed an error of 0.42–8.00% compared to the intended moisture content, but it was confirmed that the sample was prepared with an increase in moisture content with an R^2^ of 0.99 or more. The capsaicinoid content of the red pepper powders is listed in Table 1, indicating that the capsaicin content of all samples was higher than the dihydrocapsaicin content. The pungent substances in red pepper are capsaicin homologs, and the main components of capsaicinoids are capsaicin, dihydrocapsaicin, and nodihydrocapsaicin, each at approximately 70%, 21–40%, and 2–12% composition, respectively [25]. For total capsaicinoid content, GC ranged from 156.80–165.57 mg/kg, ST ranged from 252.14–269.10 mg/kg, KC ranged from 510.44–544.65 mg/kg, and JU ranged from 676.04–731.92 mg/kg. According to the Korean Industrial Standard, GC and ST are classified as ‘Slight Hot’ and KC and JU as ‘Medium Hot’. There was a slight difference in the capsaicin, dihydrocapsaicin, and total capsaicinoid content depending on the water content, but no significant differences were observed.

Park et al. [26] and Choi et al. [3] stated that fructose and glucose account for 70% of the total sugars in red pepper, and the sweetness of red pepper is in the order of fructose, glucose, and sucrose. All red pepper powders were composed of free sugars in the order of fructose > glucose > sucrose content, and the free sugar content was not affected by the water or capsaicinoid content of red pepper powder.

The American Spice Trade Association (ASTA) color values were calculated as 83.90–86.92 for JU, 75.95–79.14 for GC, 62.93–65.75, ST, and 57.72–59.65 for KC. JU, GC, ST, and KC were dark red. The ASTA color, which is a criterion for the color of red pepper powder [2] and the pigment content of red pepper powder are known to fluctuate depending on the variety, cultivation area, and drying method, such as sun drying and hot air drying [27,28,29]. Therefore, it is difficult to determine the degree of spiciness and sweetness by observing the appearance of red pepper powder with the naked eye without analysis.

The pungency components, including capsaicin, dihydrocapsaicin, and capsaicinoid, showed a low correlation with moisture content, ASTA value, and free sugars (fructose, glucose, sucrose, and total free sugar) indicating that there was no significant effect on pungency level. Therefore, when predicting the pungency level of red pepper powder using spectral information, it is proven that pungency components can show independent spectral characteristics without mutual influence between physicochemical characteristics.

### 3.2. Spectral Characteristics 

Figure 2 shows the hyperspectral mean spectra obtained from the GC, KC, ST, and JU red pepper powders with different pungency levels and moisture contents. Red pepper powder is composed of 50–60% carbohydrates, 10–15% crude protein, 10% crude fat, and 5% ash [30]. Therefore, as a result of observing the spectra, the shapes of all spectra were similar, except for the difference in the overall reflection intensity depending on the sample. In the observation of the characteristics of the average reflectance spectrum in the VIS-NIR region without any chemometrics analysis (Figure 2A), the reflectance intensity was relatively low in the sample with high moisture content, whereas differences in reflectance by pungency level, ASTA color, and free sugar were not observed.

Red pepper powder absorbs light at approximately 1130, 1200, 1425–1440, and 1515 nm in the SWIR band (Figure 2B), which is similar to the results reported by Mo et al. [4]. Each peak represents the 2nd overtone region of the CH bond (1200 nm), 1st overtone combination of CH and OH bonds (1425 nm) and 1st overtone of the NH bond (1520 nm), respectively [31,32,33,34].

In the band of approximately 1410–1540 nm, which is common in GC, ST, KC, and JU, the reflectance intensity was low in samples with high moisture content, and it seems that the absorption phenomenon was strengthened by a large number of OH bonds. However, since it is difficult to quantify the sweetness and spiciness of red pepper only by observing the average spectrum, additional chemometrics analysis is required. Therefore, by attempting multivariate analysis of hyperspectral data, there is a possibility of evaluating the quality of red pepper powder and expressing it numerically.

### 3.3. Chemometrics

#### 3.3.1. Principal Component Analysis

The original reflectance spectral data matrix was reduced to a system of coordinate axes, where samples were located according to principal component analysis (PCA) scores instead of intensities in the wavelength space [35]. Therefore, samples with similar spectral properties tend to project to the same location in principal component space. A clear differentiation according to capsaicinoid content and moisture content is indicated in the PCA score plots shown in Figure 3, which are expressed in two dimensions and three dimensions by the principal component factors based on the hyperspectral spectra. In the score plot, GC is shown in blue, ST in green, KC in yellow, and JU in orange; the higher the moisture content, the darker the color. PC−1, PC−2, and PC−3 contributed 95%, 3%, and 1% of the hyperspectral image data of red pepper powder obtained in the VIS-NIR region, respectively (Figure 3A,B). As indicated by the dotted circle, it is clearly classified according to the production area of red pepper powder, which may mean that it is classified according to the degree of spiciness or ASTA color; therefore, additional interpretation is needed through the loading plot result. In addition, the distribution of darker markers closer to the upper left corner of the score plot indicates that PCA analysis using hyperspectral data in the VIS−NIR region can visually show the difference in the moisture content of red pepper powder.

PCA results of the SWIR region showed that the first principal component (PC1) and the second principal component (PC2) accounted for 91% and 6% of the spectral variance, respectively. Because the first two principal components can explain 97% of the data, this data reveals the high feasibility of discrimination among red pepper powders. In the two-dimensional plot, it was sequentially distributed according to the moisture content, which can be the basis for the hyperspectral spectrum to represent the relative moisture content distribution of red pepper powder. In the three-dimensional plot, separate grouping was performed according to the sample and moisture content. Therefore, PCA analysis using hyperspectral data in the SWIR region can be a method that can effectively show the difference in the distribution of moisture content and other quality characteristics of red pepper powder. This plot only demonstrates the qualitative differences between the examined samples without referring to their quantitative attributes [35].

#### 3.3.2. Loading Plot

The first two PCs accounted for 97% or more of the spectral variation in the tested samples; therefore, these five PCs can be used as alternatives to the variables for the classification of red pepper powder (Figure 4). In this study, to identify the key wavelengths that are highly correlated with each PC for VIS−NIR and SWIR systems, the PC loadings were plotted against their spectral ranges, and all characteristic wavelengths were marked. PC loading can be used to identify wavelengths highly correlated with each PC [36]. In addition, the PCA results of the spectral data of all tested red pepper powder spectra loadings are the regression coefficients for each wavelength in each principal component, indicating which wavelength has a dominant effect on identification.

As a result of observing the PCA loading plot of VIS-NIR data (Figure 4A), PC1 explained 95% of the total variance in the samples. Key wavelengths (675–760 nm) were shown from this component, and key peaks were observed in the 580–610 nm, 675 nm, and 870–970 nm bands from PC2. Among the key wavelengths (580–610, 675–760, 870–975 nm) shown by PCA loadings, a peak observed in the red region (675 nm) might also be related to the presence of carotenoids [37]. The high absorbance observed at 625–740 nm is associated with red absorbing pigments, mainly chlorophyll absorption [38,39]. Absorption at 750 and 974 nm is due to water absorption bands related to O–H stretching second overtones [40,41]. Owing to the obvious difference in ASTA color value and moisture content between the samples in Table 1, VIS–NIR spectroscopic images can be used to compare the moisture content and color of red pepper powder.

As a result of observing the PCA loading plot of the SWIR data, PC–1 showed a prominent peak only at 1460 nm, and PC-2 showed peaks at 1020–1130 nm and 1430–1520 nm (Figure 4B). Capsaicin and dihydrocapsaicin are alkaloids with molecular formulas of C_18_H_27_NO_3_ and C_18_H_29_NO_3_, respectively, and the capsaicin molecule can be divided into three regions: aromatic rings, amide bonds, and hydrophobic side chains [42]. The chemical bonds that are read include O–H str. 1st overtone was detected in the wavelength range of 1395–1452 nm and this chemical bond in the form of the C–H stretch 1st overtone is due to the presence of aromatic and alkene functional groups, which are also known to be constituents of capsaicin [34]. The 2nd overtone occurred because of the presence of a hydroxyl group (-OH) derived from several sources of antioxidants in red chili, such as capsanthin and capsaicin.

Therefore, it is foreseen that wavelengths at water absorption bands and capsaicinoid absorption bands are important for discrimination of pungency level and moisture content within each red pepper powder.

#### 3.3.3. Prediction of Quality Attribute in Red Pepper Powder

The prediction results of capsaicinoid, free sugar, and ASTA color by PLS modeling in VIS-NIR and SWIR are shown in Figure 5 and Figure 6. The average R_p_^2^ of Model A in VIS-NIR for capsaicinoid was 0.98, and the average R^2^ value decreased to approximately 0.92 in B7-10, B7-11 and B7-12 models, respectively: A decrease in R_p_^2^ of approximately 5.9% occurred. The SWIR R_p_^2^ values of the B7-10, B7-11, and B7-12 prediction models for the capsaicinoid were 0.85–0.87, a decrease of approximately 8.7% from the average R_p_^2^ value of A7–A12. Referring to Figure 4A, the loading peaks at 590 nm and 670 nm, which can explain the red color, were about 0.04 higher than those at 750 nm and 970 nm related to moisture. On the other hands, there is a peak that stands out more than other bands at 1450 nm where the vibration of OH bond in water molecules is revealed in Figure 4B. Therefore, the SWIR spectra were more sensitive to the moisture content of the sample compared to VIS-NIR spectra, which hindered the prediction of capsaicinoid content by difference of water contents.

The modeling results for free sugars are as follows. In Figure 5, the prediction Model A with uniform moisture content had an R_p_^2^ value of 0.96 or more. However, R_p_^2^ decreased in the order of B7-8 (0.94), B7-9 (0.90), B7-10 (0.90), B7-11 (0.85), and B7-12 (0.80) models. In Figure 6, it can be observed that the average R_p_^2^ of Model A is 0.951, whereas that of Model B is 0.839, a decrease of about 12%. As shown in Figure 5, the fact that the R_p_^2^ value did not decrease sequentially can be interpreted as a slight error according to the resolution of the SWIR system itself and the number of measurement bands. As a result, it means that the adjustment of the water content of the sample has a significant effect on the accuracy of the PLS model in predicting the free sugar content in both the VIS-NIR and SWIR regions.

The training, and prediction model of the ASTA color value in VIS-NIR maintained an R_c_^2^, R_cv_^2^ and R_p_^2^ of 0.97 or more regardless of the moisture content distribution. In the SWIR region, it was observed that the R^2^ values of the B7-11 and B7-12 models slightly decreased below 0.95 in the ASTA prediction model, but the prediction accuracy was still high. Although capsanthin, zeaxanthin, cryptoxanthin, and betacarotene are responsible for the red color in red pepper powders [43], the use of VIS-NIR region, which was based on the external color values of red peppers was better for developing the prediction model of ASTA color value than the use of SWIR region, which was based on the chemical structure of red peppers by water molecules (OH bond). Therefore, the hyperspectral imaging system is more useful and convenient for estimating ASTA values because there is less need to adjust the moisture content of the sample.

## 4. Conclusions

The present study predicted the capsaicinoid and free sugar content through hyperspectral imaging and PLS analysis of red pepper powder with different moisture contents and different pungency levels. There is an explicit tendency for the RMSE value to increase as the difference in moisture content of the modeling sample increases for all predicted quality attributes. Finally, a difference of more than 2% in MC had a negative effect on prediction accuracy for capsaicinoid and free sugar. Therefore, this study demonstrated that it is essential to adjust the moisture content difference of red pepper powder samples to be used for modeling within 2% using a hyperspectral imaging system. It is expected that this will be used as a basis for the development of automated systems for the rapid grading of pungency levels and sweetness.

## Figures and Tables

**Figure 1 foods-11-04086-f001:**
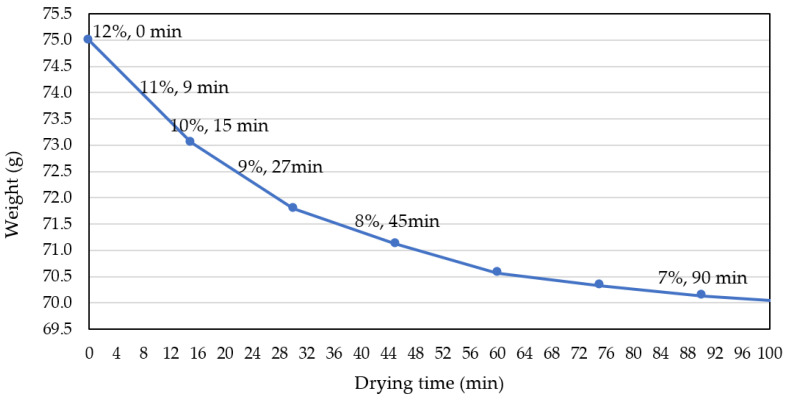
Weight change of red pepper powder according to drying time and calculated moisture content.

**Figure 2 foods-11-04086-f002:**
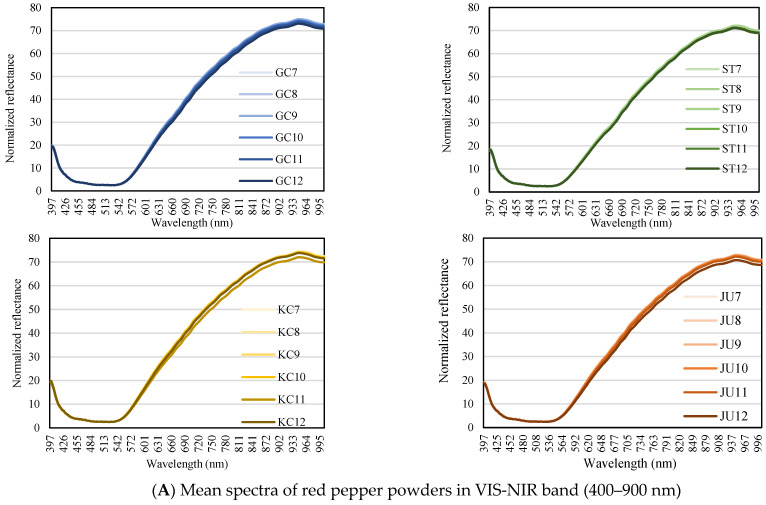
Mean spectra of red pepper powders in the Vis-NIR (**A**) and SWIR (**B**) wavelength ranges according to pungency levels and moisture contents.

**Figure 3 foods-11-04086-f003:**
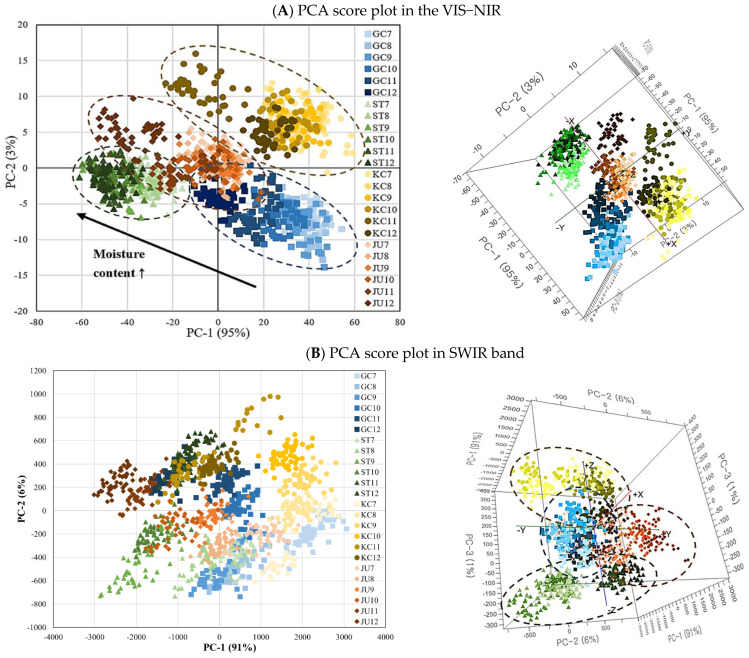
PCA score plot of hyperspectral spectra in the VIS−NIR (**A**) and SWIR band (**B**).

**Figure 4 foods-11-04086-f004:**
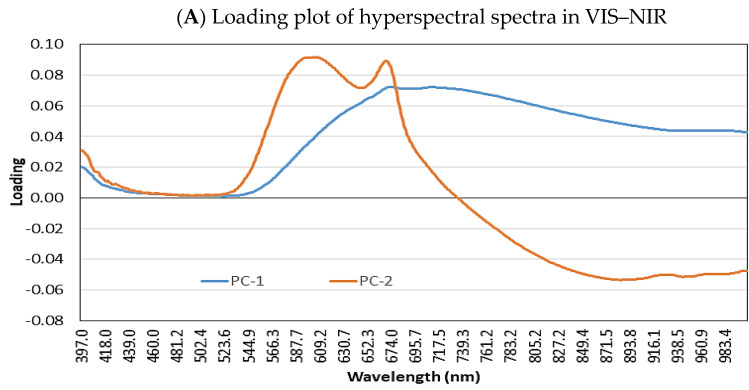
Loading plot of PC1 and PC2 derived from PCA of hyperspectral spectra in VIS–NIR (**A**) and SWIR band (**B**).

**Figure 5 foods-11-04086-f005:**
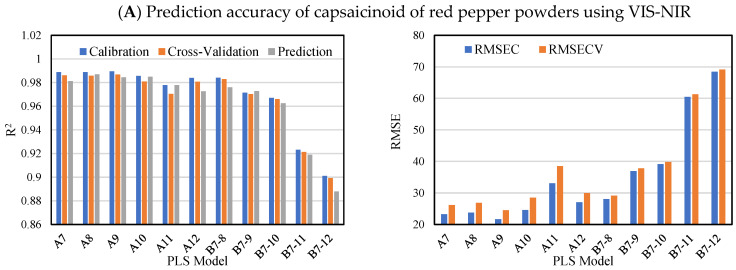
Prediction accuracy of capsaicinoid (**A**), free sugar (**B**) and ASTA color (**C**) of red pepper powders using VIS-NIR wavelength range in accordance with moisture content. RMSEC, root mean square error of calibration; RMSECV, root mean square error of cross-validation.

**Figure 6 foods-11-04086-f006:**
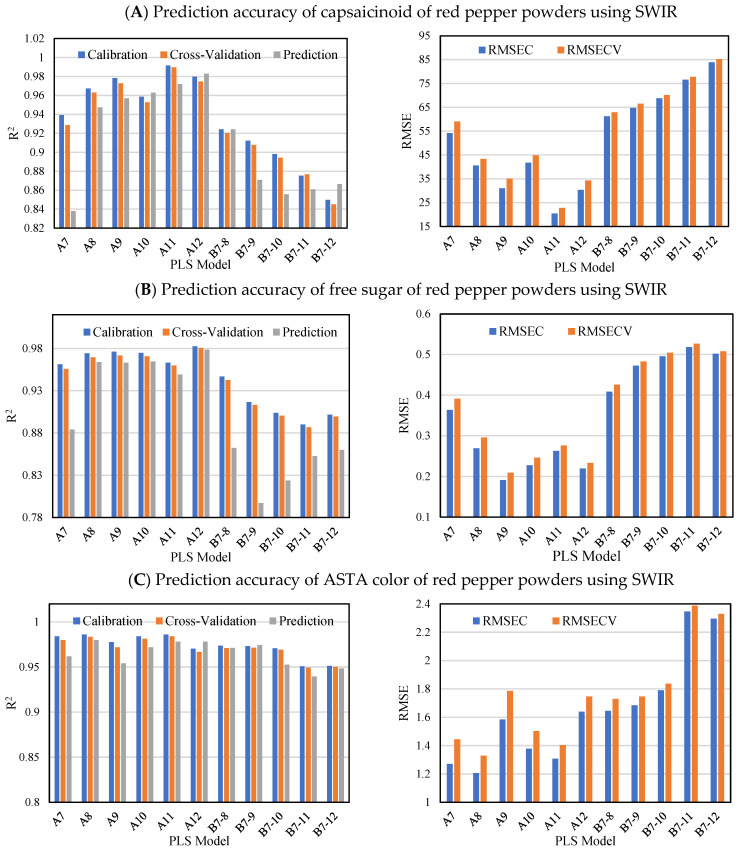
Prediction accuracy of capsaicinoid (**A**), free sugar (**B**) and ASTA color (**C**) of red pepper powders using SWIR wavelength range in accordance with moisture content. RMSEC, root mean square error of calibration; RMSECV, root mean square error of cross-validation.

**Table 1 foods-11-04086-t001:** Developed partial least square model.

Model Name		N	Inserted Data
Model A	A7	200	GC7, ST7, KC7, JU7
	A8	200	GC8, ST8, KC8, JU8
	A9	200	GC9, ST9, KC9, JU9
	A10	200	GC10, ST10, KC10, JU10
	A11	200	GC11, ST11, KC11, JU11
	A12	200	GC12, ST12, KC12, JU12
Model B	B7-8	400	GC7, ST7, KC7, JU7, GC8, ST8, KC8, JU8
	B7-9	600	GC7, ST7, KC7, JU7, GC8, ST8, KC8, JU8, GC9, ST9, KC9, JU9
	B7-10	800	GC7, ST7, KC7, JU7, GC8, ST8, KC8, JU8, GC9, ST9, KC9, JU9, GC10, ST10, KC10, JU10
	B7-11	1000	GC7, ST7, KC7, JU7, GC8, ST8, KC8, JU8, GC9, ST9, KC9, JU9, GC10, ST10, KC10, JU10, GC11, ST11, KC11, JU11
	B7-12	1200	GC7, ST7, KC7, JU7, GC8, ST8, KC8, JU8, GC9, ST9, KC9, JU9, GC10, ST10, KC10, JU10, GC11, ST11, KC11, JU11, GC12, ST12, KC12, JU12

**Table 2 foods-11-04086-t002:** Physicochemical properties of red pepper powder according to pungency level and moisture contents.

Sample ^(1)^	Moisture Content (%)	ASTA Value	Capsaicinoid (mg/kg)			Free Sugar (%)			
			Capsaicin	Dihydrocapsaicin	Total	Fructose	Glucose	Sucrose	Total
GC7	7.24 ± 0.06 ^b(2)^	77.33 ± 0.16 ^f^	98.04 ± 2.66 ^a^	64.84 ± 1.37 ^a^	162.89 ± 4.00 ^a^	6.86 ± 0.02 ^ab^	3.19 ± 0.08 ^ab^	0.95 ± 0.02 ^i^	11.00 ± 0.12 ^ab^
GC8	8.29 ± 0.05 ^e^	76.37 ± 0.88 ^f^	96.84 ± 2.65 ^a^	63.62 ± 0.97 ^a^	160.46 ± 3.59 ^a^	6.88 ± 0.04 ^ab^	3.17 ± 0.09 ^ab^	0.98 ± 0.01 ^ij^	11.03 ± 0.09 ^ab^
GC9	9.35 ± 0.05 ^h^	77.27 ± 0.88 ^f^	99.30 ± 0.65 ^a^	66.26 ± 0.38 ^a^	165.57 ± 1.03 ^a^	7.09 ± 0.05 ^bc^	3.23 ± 0.05 ^b^	1.00 ± 0.02 ^jk^	11.32 ± 0.06 ^b^
GC10	10.39 ± 0.04 ^k^	77.25 ± 0.31 ^f^	94.41 ± 1.02 ^a^	63.58 ± 0.28 ^a^	158.99 ± 1.17 ^a^	6.92 ± 0.06 ^ab^	3.10 ± 0.04 ^ab^	1.01 ± 0.01 ^jk^	11.03 ± 0.09 ^ab^
GC11	11.54 ± 0.05 ^m^	79.14 ± 0.44 ^g^	94.35 ± 2.07 ^a^	63.85 ± 1.76 ^a^	158.20 ± 3.80 ^a^	6.95 ± 0.25 ^ab^	3.03 ± 0.16 ^ab^	0.99 ± 0.03 ^jk^	10.97 ± 0.44 ^ab^
GC12	12.35 ± 0.08 ^p^	75.95 ± 2.99 ^f^	93.51 ± 0.63 ^a^	63.28 ± 0.76 ^a^	156.80 ± 1.26 ^a^	6.72 ± 0.03 ^a^	2.98 ± 0.04 ^ab^	1.00 ± 0.02 ^jk^	10.69 ± 0.09 ^a^
ST7	7.13 ± 0.02 ^a^	64.52 ± 0.19 ^de^	159.22 ± 2.17 ^d^	105.91 ± 1.79 ^bc^	265.13 ± 3.96 ^cd^	9.09 ± 0.14 ^ij^	5.11 ± 0.08 ^j^	0.62 ± 0.03 ^a^	14.82 ± 0.24 ^hi^
ST8	8.16 ± 0.02 ^d^	64.39 ± 0.63 ^de^	160.98 ± 4.65 ^d^	108.13 ± 3.55 ^c^	269.10 ± 8.16 ^d^	9.14 ± 0.14 ^ij^	5.13 ± 0.06 ^j^	0.66 ± 0.02 ^bc^	14.94 ± 0.21 ^hi^
ST9	9.41 ± 0.04 ^h^	62.93 ± 0.48 ^c^	157.00 ± 1.57 ^cd^	105.67 ± 0.61 ^bc^	262.68 ± 2.14 ^bcd^	8.82 ± 0.15 ^efghi^	4.84 ± 2.00 ^ghi^	0.64 ± 0.01 ^ab^	14.30 ± 0.25 ^defgh^
ST10	10.34 ± 0.05 ^k^	63.49 ± 0.42 ^cd^	159.82 ± 0.25 ^d^	108.12 ± 0.36 ^c^	267.94 ± 0.56 ^d^	9.15 ± 0.21 ^ij^	5.00 ± 1.57 ^ij^	0.68 ± 0.01 ^c^	14.83 ± 0.28 ^hi^
ST11	11.30 ± 0.06 ^l^	65.75 ± 0.96 ^e^	149.91 ± 3.58 ^b^	102.24 ± 2.39 ^b^	252.14 ± 5.97 ^b^	8.96 ± 0.12 ^hij^	4.82 ± 1.59 ^fghi^	0.67 ± 0.03 ^bc^	14.46 ± 0.22 ^fghi^
ST12	12.05 ± 0.02 ^o^	65.30 ± 0.41 ^e^	150.40 ± 1.74 ^bc^	102.73 ± 1.38 ^bc^	253.13 ± 3.12 ^bc^	9.17 ± 0.13 ^ij^	4.93 ± 1.92 ^hij^	0.68 ± 0.01 ^c^	14.79 ± 0.22 ^hi^
KC7	7.56 ± 0.13 ^c^	59.91 ± 0.71 ^b^	280.00 ± 5.44 ^g^	251.80 ± 5.38 ^fg^	531.80 ± 10.81 ^fg^	9.79 ± 0.10 ^k^	5.39 ± 0.08 ^k^	0.74 ± 0.01 ^e^	15.92 ± 0.17 ^j^
KC8	8.64 ± 0.06 ^f^	59.65 ± 0.38 ^b^	285.14 ± 3.05 ^gh^	257.28 ± 2.30 ^hi^	542.42 ± 5.34 ^gh^	9.27 ± 0.41 ^j^	5.06 ± 0.24 ^j^	0.74 ± 0.04 ^de^	15.07 ± 0.69 ^i^
KC9	9.18 ± 0.02 ^g^	59.45 ± 0.38 ^b^	286.09 ± 3.84 ^h^	258.56 ± 3.00 ^hi^	544.65 ± 6.84 ^h^	8.55 ± 0.13 ^def^	4.59 ± 0.08 ^de^	0.70 ± 0.01 ^cd^	13.84 ± 0.21 ^de^
KC10	10.19 ± 0.04 ^j^	59.50 ± 0.36 ^b^	278.72 ± 10.04 ^fg^	251.81 ± 9.08 ^fg^	530.53 ± 19.12 ^fg^	8.58 ± 0.06 ^defg^	4.57 ± 0.02 ^de^	0.71 ± 0.01 ^cde^	13.87 ± 0.06 ^def^
KC11	11.37 ± 0.03 ^l^	57.72 ± 0.35 ^a^	267.65 ± 4.85 ^e^	242.79 ± 3.87 ^d^	510.44 ± 8.72 ^e^	8.53 ± 0.08 ^de^	4.44 ± 0.05 ^d^	0.74 ± 0.01 ^e^	13.72 ± 0.13 ^d^
KC12	12.08 ± 0.04 ^o^	59.27 ± 0.37 ^b^	272.20 ± 2.01 ^ef^	247.76 ± 1.69 ^def^	519.96 ± 3.70 ^ef^	8.91 ± 0.05 ^fghij^	4.65 ± 0.05 ^defg^	0.79 ± 0.02 ^f^	14.35 ± 0.12 ^efgh^
JU7	7.54 ± 0.03 ^c^	85.24 ± 0.49 ^hi^	459.26 ± 5.14 ^k^	262.04 ± 2.81 ^ij^	721.30 ± 7.95 ^k^	8.83 ± 0.09 ^efghi^	4.74 ± 0.06 ^efgh^	1.01 ± 0.03 ^jk^	14.58 ± 0.16 ^ghi^
JU8	8.36 ± 0.02 ^e^	85.66 ± 0.44 ^ij^	465.90 ± 5.59 ^k^	266.02 ± 3.13 ^j^	731.92 ± 8.71 ^k^	8.93 ± 0.03 ^ghij^	4.77 ± 0.06 ^efgh^	1.03 ± 0.01 ^k^	14.73 ± 0.08 ^hi^
JU9	9.56 ± 0.04 ^i^	85.47 ± 0.98 ^i^	438.91 ± 2.58 ^j^	250.88 ± 1.95 ^efg^	689.79 ± 4.53 ^j^	8.69 ± 0.29 ^defgh^	4.58 ± 0.22 ^de^	1.01 ± 0.04 ^jk^	14.29 ± 0.53 ^defgh^
JU10	10.24 ± 0.03 ^j^	86.92 ± 0.64 ^j^	430.31 ± 3.54 ^i^	245.73 ± 1.82 ^de^	676.04 ± 5.37 ^i^	8.43 ± 0.56 ^d^	4.61 ± 0.33 ^def^	0.92 ± 0.01 ^h^	13.96 ± 0.90 ^defg^
JU11	11.66 ± 0.03 ^n^	85.95 ± 0.67 ^ij^	461.74 ± 6.55 ^k^	264.00 ± 3.25 ^j^	725.75 ± 9.79 ^k^	8.55 ± 0.21 ^def^	4.46 ± 0.11 ^d^	1.03 ± 0.01 ^k^	14.04 ± 0.28 ^defg^
JU12	12.55 ± 0.08 ^q^	83.90 ± 0.38 ^h^	443.85 ± 5.30 ^j^	253.51 ± 2.77 ^gh^	697.36 ± 8.07 ^j^	7.40 ± 0.19 ^c^	3.82 ± 0.09 ^c^	0.88 ± 0.03 ^g^	12.10 ± 0.26 ^c^

^(1)^ GC, red pepper powder produced in Gochang-gun; ST, red pepper powder produced in Sintaein-eup; KC, red pepper powder produced in Kwanchon-myeon; JU, red pepper powder produced in Jeongeup-si. ^(2)^ Mean ± standard deviation (*n* = 3) with different superscript letters is significantly different at 5% level.

## Data Availability

Not applicable.

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
