# Peer review of "Effect of Moisture Content Difference on the Analysis of Quality Attributes of Red Pepper (Capsicum annuum L.) Powder Using a Hyperspectral System"

_foods, 2022, doi:10.3390/foods11244086_

Round 1

Reviewer 1 Report

The manuscript presents deals with the application of hyperspectral imaging in predicting the quality of red pepper powder by adjusting moisture content. The hypothesis is clear and well-explained. The language of the manuscript is clear and easy to understand. The authors have used proper statistical design to predict the quality of red pepper powder.

I have the following suggestions-

       i.          Introduction: This section provides sufficient background information.

     ii.          Keywords: appropriate

   iii.          L58: please mention the trends in the place described positively

   iv.          L71-73: please reframe sentence

     v.          L90-91: Is there any criteria set for sampling, if yes then please specify

   vi.          L163: please mention n?

  vii.          Results: well explained by suitable references

viii.          Conclusion: missing, may be the para with line L426-431 fit well for the same. 

Author Response

Thank you very much for taking the time to review my paper. As you suggested, I did my best to revise it as follows. All modified parts are marked in red. Please advise again on the modified part that is different from your intention. thank you

1. L58: please mention the trends in the place described positively

Unnecessary parts of sentences have been deleted.

2.  L71-73: please reframe sentence

Sentences were reframed as indicated in L76-78.

3. L90-91: Is there any criteria set for sampling, if yes then please specify

Korean Industrial standards for red pepper powder were explained in L97-103. 

4. L163: please mention n?

In L174-L175, the number of samples and n number of measurements are mentioned.

5.  Conclusion: missing, may be the para with line L426-431 fit well for the same. 

In Section 4. Conclusion

The conclusion part was prepared using the existing paragraphs of L 426-431.

Reviewer 2 Report

I reviewed the manuscript entitled, Effect of moisture content difference on the analysis of quality attributes of red pepper (Capsicum annuum L.) powder using a hyperspectral system. This work is novel and contributes to the field. Authors should consider below suggestions

Abstract: Please introduce the background of the study

Introduction

Authors should introduce the hyperspectral system and approach. The use of PCA and its application in food should be addressed

What authors considered moisture content?

Line 91: drying.. What is the drying temperature?

Lines 97-102: please provide reference

Figure 1. quality should be improved

Please provide reference for 2.2.2. Capsaicinoid

Please provide ref for 2.2.3. Free sugar

Results and discussion are appropriate and supported with available literature.

Please provide conclusion section

References are not according to the journal. Please revise it 

Author Response

Thank you very much for taking the time to review my paper. As you suggested, I did my best to revise it as follows. All modified parts are marked in red.Please advise again on the modified part that is different from your intention. thank you

1. Abstract: Please introduce the background of the study

L12-13,  "The variety of characteristics of red pepper makes it difficult to analyze at the production field through hyperspectral imaging"

2. The use of PCA and its application in food should be addressed

L55-60, Background and references to the use of PCA in food analysis were added.

3. What authors considered moisture content?

L101-103, A background setting the water content of the sample was added.

"KS presents less than 13% as the appropriate moisture content of red pepper powder, and the average moisture content of red pepper powder sold on the market is 7-12%."

4. Line 91: drying.. What is the drying temperature?

L95, The drying temperature was 50-60 °C.

5. Figure 1. quality should be improved

The resolution of fig 1 has been improved.

6. Please provide reference for 2.2.2. Capsaicinoid and 2.2.3. Free sugar

A reference for the assay was provided.

7. Please provide conclusion section

L440-449, A conclusion section has been added.

8. References are not according to the journal. Please revise it 

References have been corrected according to the journal.
